# Investigation of Growth Mechanism of Plasma Electrolytic Oxidation Coating on Al-Ti Double-Layer Composite Plate

**DOI:** 10.3390/ma12020272

**Published:** 2019-01-15

**Authors:** Quanzhi Chen, Weizhou Li, Kui Ling, Ruixia Yang

**Affiliations:** 1School of Resources, Environment and Materials, Guangxi University, Nanning 530004, China; quanzhi_chen@163.com (Q.C.); 1815301022@st.gxu.edu.cn (K.L.); yrx20070018@gxu.edn.cn (R.Y.); 2Guangxi Key Laboratory of Processing for Non-ferrous Metals and Featured Materials, Guangxi University, Nanning 530004, China

**Keywords:** Al-Ti double-layer composite plate, plasma electrolytic oxidation, growth mechanism

## Abstract

The aluminum–titanium (Al-Ti) double-layer composite plate is a promising composite material, but necessary surface protection was required before its application. In this paper, plasma electrolytic oxidation (PEO) was employed to fabricate a ceramic coating on the surface of a Al-Ti double-layer composite plate. To investigate the coating growth mechanism on the Al-Ti double-layer composite plate, a single-Al plate and a single-Ti plate were introduced for comparison experiments. Results showed that, the composite of Al and Ti accelerated the coating growth rate on the part-Ti portion of the composite plate, and that of the part-Al portion was decreased. Electrochemical impedance spectroscopy analysis indicated that the equivalent circuit of the Al-Ti coating was formed by connecting two different circuits in parallel. The reaction behavior revealed that the electric energy during the PEO would leak from the circuit with the weaker blocking effect, and confirmed that the electric energy distribution followed the law of low-resistance distribution. Finally, the mechanism was extended to the PEO treatment on general metal matrix composites to broaden the application theory of the technology.

## 1. Introduction

The aluminum–titanium (Al-Ti) double-layer composite plate prepared by explosive welding technology has been used widely in aviation, shipbuilding, petrochemical, and other fields, which is due to the excellent corrosion resistance and mechanical properties of the Ti alloy, as well as the high specific strength, low cost, and nice formation property of the Al alloy [1]. The combination of two heterogeneous metals, however, inevitably leads to galvanic corrosion [2] and other hidden dangers at the welding joint. Therefore, surface protection treatment is required before its application. There are many surface protection techniques, including anodic oxidation [3], plasma electrolytic oxidation (PEO) [4], and plasma spraying [5]. Among these techniques, PEO is an effective surface modification technique for fabricating ceramic oxide coatings on light metals, which has been widely applied to Al, Ti, and other valve alloys [6,7]. Therefore, it could be used to prepare a protective coating on the surface of Al-Ti composite plate and welding joint simultaneously.

At present, much research on the PEO of metal matrix composites (MMCs) has been reported. For example, Xue [8] conducted PEO treatment on Al/SiCP composites and found that the SiC particles would become molten and then oxidized to become a silicon oxide coating. Xu et al. [9] studied the growth of PEO coating on Al-15.6%Si alloy and found that puncture discharge occurred preferentially at the interface of Al/Si under the action of tip discharge, and had a significant effect on the discharge mechanism and coating structure. It could be seen that the heterogeneous phase would consume electrical energy and reduce its conversion efficiency, and then ultimately affect the coating structure and performance. These studies, however, tracked and observed only the changes of the heterogeneous phase, and no studies have been conducted on the conversion of electrical energy during the reaction.

PEO is an electrochemical reaction process that converts electrical energy into chemical energy. The conversion efficiency of electrical energy has a great influence on the coating structure and properties. Therefore, the combination of coating structure and electrical energy changes provides a new research direction to study the PEO mechanism. In this paper, we investigated the growth law of PEO coating on an Al-Ti double-layer composite plate from the perspectives of electric energy conversion and coating structure. In order to study the reaction behavior accurately, single-Al and single-Ti alloys were introduced for comparative experiments. Finally, the mechanism was extended to general MMCs to broaden the application theory of the PEO technology.

## 2. Materials and Methods

Single-Al alloy, single-Ti alloy, and an Al-Ti double-layer composite plate prepared by explosive welding was used as substrates in this experiment. The chemical composition of the Al alloy was (wt.%) 0.25 Fe, 0.20 Si, 0.015 Cu, and balanced Al; the chemical compositions of the Ti alloy was 0.30 Fe, 0.10 C, 0.15 Si, 0.05 N, and balanced Ti. The dimensions of all substrates were 20 mm × 15 mm × 3 mm. The Al-Ti composite plate, with the 3 mm thickness being 1.5 mm Al and 1.5 mm Ti, and its diagrammatic drawing is shown in Figure 1. For convenience, single-Al alloy, single-Ti alloy, part-Al, and part-Ti were referred as S-Al, S-Ti, P-Al, and P-Ti, respectively. Prior to PEO processing, the substrate was polished with 400^#^, 600^#^, 1000^#^, and 1500^#^ grit silicon carbide papers, and then was cleaned with acetone and distilled water.

A pulse power supply (YISHENG Electronics Science & Technology Co, LTD, Shanghai, China) was employed for the PEO treatment. The electrolyte consisted of Na_2_SiO_3_ (15 g/L), NaF (2 g/L), and Na(PO_3_)_6_ (5 g/L). Two experiments were made: One was the reaction behavior study, which was conducted at a constant current mode with a current density, frequency, duty ratio, and processing time of 10 A/dm^2^, 500 Hz, 15%, and 20 min, respectively. The second experiment studied the structure of the coating, which was done at constant voltage mode, with the voltage, frequency, and duty ratio set as 470 V, 500 Hz, and 15%, respectively. The processing times were 5, 10, 20, 30, 40, 50, 60, 90 min. The electrolyte temperature was kept below 40 °C with a water bath during the PEO process.

The coating surface and cross-section morphologies were observed by scanning electron microscopy (SEM, Hitachi S-3400N, Hitachi SU-8020, Tokyo, Japan). Surface roughness measurement was done by atomic force microscopy (AFM, Hitachi 5100N, Tokyo, Japan), and the roughness analysis was carried out using the analysis software provided by the AFM equipment. We calculated the average and standard deviation after taking the roughness measurements in three different regions on the topographic map. The phase structure was determined by X-ray diffraction (XRD, Cu Kα radiation, Malvern, UK) using a PANalytical X’Pert-PRO instrument with a step size of 0.026° and an energy of 40 kV and 40 mA. The coating thickness was measured by a TT260B Coating Thickness Gauge (SDCH Co., LTD, Beijing, China). The measurement was performed 10 times at different places on every sample, and the average value and its standard deviation were determined. The electrochemical impedance spectroscopy (EIS) was conducted by an electrochemical workstation CHI 750e (CH Instruments, Inc, Shanghai, China), and corrosive medium was 3.5 wt.% sodium chloride solution. A saturated calomel electrode (SCE) was used as a reference electrode, and a platinum plate as the counter electrode. The EIS was performed at frequencies between 0.1 Hz and 100 kHz, and amplitude of the sinusoidal voltage signal was 10 mV. Before the test, the P-Al and P-Ti coating surfaces were coated with paraffin, and a working area of ~1cm^2^ was left at the welding zone. 

## 3. Results and Discussion

### 3.1. Phase Structure and Morphology Analysis

The XRD patterns of the different PEO coatings (Figure 2a,b) demonstrate that the phase structures of the coatings of S-Al and P-Al were composed mainly of γ-Al_2_O_3_. Similarly, the phase structures of the S-Ti and P-Ti coatings were the same, mainly containing rutile TiO_2_ (R-TiO_2_) and anatase TiO_2_ (A-TiO_2_). These results suggested that the composite of Al and Ti did not change the phase structure because the P-Al and P-Ti of the Al-Ti double-layer composite plate were isolated and had no effect on the coating phase structure.

In order to study the growth law of coating, the surface morphology and roughness of different coatings with different treatment times were analyzed, as shown Figure 2c–f. It is evident that the surface roughness of different samples increased with an increasing treatment time. Figure 2c shows that the roughness of the P-Ti coating was higher than that of the S-Ti coating throughout the experiment; the roughness of the P-Ti coating at 60 min was 1.37 μm, and that of the S-Ti coating was 1.22 μm. Combined with the surface morphology analysis, the surface of the S-Ti and P-Ti coatings were relatively smooth at 10min, and the discharge pores were also small, as shown in Figure 2e. At longer experimental times, the discharge pore size increased and formed discharge pits. Additionally, the discharge pits in the P-Ti coating were deeper than that of the S-Ti coating, as shown in the insets of Figure 2c and Figure 2f. As the electric energy increased, the reaction intensity grew stronger, so that the discharge pits formed during the breakdown process were increased, thereby obtaining a coating with higher roughness. This revealed that the electric energy distributed on the P-Ti surface was higher than that of S-Ti, and the breakdown discharge intensity was higher. A comparative analysis of the P-Al and S-Al coatings surface roughness (Figure 2d) showed that the surface roughness of both coatings increased before 40 min, but that of S-Al decreased slightly at 60 min. The morphology analysis (Figure 2e) illustrated that the volcano-like micropores were randomly distributed on the coating surface that was treated 10 min. When the treatment time was 60 min, however, the micropores on the S-Al coating exhibited a plugging phenomenon, shown as point A in Figure 2f. This phenomenon was not found on the P-Al coating and was due to the relatively large amount of melt ejected from the discharge channel during the breakdown process, which solidified around the pores. This reaction showed that the electric energy on the S-Al surface was higher than that of P-Al. Combined with the analysis of surface morphology and surface roughness, it was evident that the uneven distribution of electric energy occurred on the Al-Ti composite plate surface during the PEO progress.

From the cross-section morphology of the coatings treated for 60 min (Figure 2g), it could be seen that the thickness of the P-Ti coating was ~39.2 μm, whereas that of the P-Al coating was only ~8.1 μm. In addition, the P-Ti coating was thicker than the S-Ti coating, and the P-Al coating thickness was lower than that of S-Al. These results suggested that the composite of Al and Ti enhanced the reaction intensity of P-Ti while suppressing that of P-Al. The electric energy segregation that occurred on the Al-Ti composite plate during the reaction resulted in much more electric energy being concentrated on P-Ti, but the energy on the P-Al surface was lowered.

To further study the growth law of these coatings, the coating thickness and cross-section morphologies with different experimental times were characterized. The responses of thickness versus time of different samples are shown in Figure 3a,b. The coating growth process of all samples could be divided into two stages: one was the rapid growth stage when the time was between 0 and 10 min; the second was the stable growth stage when the time was between 10 and 90 min. Figure 3a illustrates that the coating thickness of S-Al was higher than that of P-Al throughout the reaction process. The final thickness of P-Al was ~13.1 μm, and that of S-Al was ~18.2 μm. It was observed that there was one obvious fluctuation in the thickness growth process of the P-Al coating, that being the growth rate increased at the period of 30–40 min. Conversely, the thickness curve of the P-Ti coating was above that of S-Ti throughout the reaction process, suggesting the growth rate of the P-Ti coating was higher than that of S-Ti. The final thickness of P-Ti was ~47.9 μm, while that of S-Ti was ~35.2 μm. During the growth process of the P-Ti coating, a slowing down of the growth rate was observed at the period of 30–40 min, which was the accelerated growth rate stage of P-Al coating. This slowing down may be due to the coating thickness having reached a certain level, with its resistance being larger than that of the P-Al coating, so that the electric energy was transferred to the P-Al coating.

Figure 3c–f show the cross-section morphologies at the welding joint at different experimental times. A coating of about ~3 μm formed on the surface of the welding joint between P-Al and P-Ti at 5 min. The coating between the two phases was well-bonded, with no cracks appearing. During the discharge breakdown process, the plasma temperature was higher than 4000 K [10], which is higher than the melting point of TiO_2_ or Al_2_O_3_. Therefore, the ejection during the breakdown process combined and then solidified into a continuous coating under water quenching. At 20 min, the thickness of P-Ti was about ~14 μm, which was ~9 μm thicker than the P-Al coating. At 40 min, the P-Ti coating thickness rapidly increased and expanded toward P-Al, as shown by the arrow in Figure 3e, but no significant change in thickness of P-Al coating was seen. At 90 min, the P-Ti coating thickness was ~30 μm larger than that of P-Al coating and expanded to P-Al at about a distance of ~28.2 μm from the wielding joint, with a step appearing in the transition region. This phenomenon confirmed that the electric energy was segregated toward the P-Ti surface during the PEO process, and accelerated its growth rate.

### 3.2. Electrochemical Performance Analysis

The EIS test was performed at the welding joint of Al-Ti coating treated for 60 min, and the results are shown in Figure 4. The Nyquist spectrum consisted of an impedance arc at the high frequency region and a diffusion arc at the low frequency region. Among them, the impedance arc at the high frequency region could be considered as a superposition of the respective capacitive reactance arcs of the two coatings, whereas the diffusion arc at low frequency was due to the porosity of the coating and diffusion at the welding joint. The Bode spectrum revealed that the coating at the welding joint possessed the characteristics of typical coating, which could also protect the substrate.

The equivalent circuit (EC) obtained by fitting the EIS spectra is shown in Figure 4b, and the fitting results are given in Table 1. These results showed that the EC was mainly formed by the parallel connection of the resistor (R), capacitance (C), and the constant phase angle elements (CEP). The EC illustrates that both P-Al and P-Ti coatings contained both inner and outer layers. The study from Chen et al. [11] showed that the EIS spectrum of the PEO coating on Al alloy possessed only one time constant, and the R values of the inner and outer layers were relatively close. Therefore, the values of R_1_ and R_2_ in Table 1 correspond to the impedance values of the inner and outer layers of the P-Al coating, respectively. Due to its high thickness and porosity, P-Ti coating could possess a higher impedance value and the strong capacitance characteristic, so that the R_5_ and C could belong to the components of the P-Ti coating EC. In addition, because of the large difference in characteristics of the P-Al and P-Ti coatings, a coupling resistance formed, and its fitting value was 1.08 × 10^3^ Ω·cm^2^, shown as R_3_. This results show that the EC of Al-Ti coating could be divided into two circuits for the P-Al and P-Ti coatings.

For visual analysis, the EC could be adjusted to the schematic shown in Figure 4d. It could be seen that the total circuit diagram was formed by connecting the ECs of the P-Al and P-Ti coatings in parallel, wherein, R_s_ was the electrolyte resistance; R_Al_ and R_Ti_ referred to the resistance of P-Al and P-Ti substrate, respectively; CEP_1_ referred to the constant phase angle elements of the P-Al coating, respectively. The CEP_2_ represent constant phase angle elements of the P-Ti coating. In addition, there is a coupling resistor (R_C_) between the circuits of the P-Al and P-Ti coatings, which was due to the large difference in the structure of the two, and an impedance could be generated under the action of the high-frequency voltage. Because the current follows the low-resistance distribution theory during the PEO process, the electrical energy in this circuit would leak from the circuit which has poor blocking effect.

In order to verify the electrical energy selective leakage problem, the reaction behaviors of S-Ti, S-Al, and Al-Ti were tested at constant current mode, and the results are shown in Figure 4c. In the response of voltage-time of different samples, three main stages were identified: rapid rise stage, transition stage, and stabilization stage. In the first stage, from 0 to ~50 s, the voltage increased linearly with a high slope. During this stage, many air bubbles and no micro-discharge were observed on the sample surface, and this stage was defined as the anodization stage. In this stage, the voltage of S-Ti increased the slowest, the S-Al was the fastest, and the Al-Ti composite plate was between the two. As is known, the increase of the voltage was caused by the increase in the resistance of the barrier layer [12]. It can be seen that the barrier layer of the Al surface has a better blocking effect on current than that of Ti. The second stage began with the emergence of discharge breakdown. Herein, the breakdown voltage of S-Ti was the lowest, at about 294 V. For S-Al, the breakdown voltage was the highest, at about 325 V, and this showed that the coating on Ti surface was more easily broken down. For the case of Al-Ti, the breakdown voltage of P-Ti was about 286 V, and that of P-Al was 331 V, which was similar to the case of the S-Ti and S-Al. In the third stage, the voltage increased slowly and smoothly. The voltage of S-Al remained at the highest value at ~470 V, and the voltage of S-Ti was the lowest at just about 380 V. For the Al-Ti composite plate, the stable phase was at about 420 V, which was between that of P-Al and P-Ti plates. This was due to the presence of the P-Al coating, which limited the leakage of the electric energy, so that the stable voltage was higher than that of S-Ti plate; and due to the presence of P-Ti coating, the electric energy was consumed, so the voltage lower than that of S-Al plate. These results confirmed that the EC of the coating on the Al-Ti coating could be connected in parallel with two different circuits and that the electric energy would select the path of the P-Ti coating, which with weaker current blocking effect, resulted in affecting the conversion efficiency of electric energy.

This rule also could be applied to general metal matrix composite (MMCs). Because there are different heterogeneous phases in the MMCs, its EC could be connected in parallel by multiple circuits, as shown in Figure 4e. On the basis of our analysis, the electric energy would preferentially leak from the circuit with poor current blocking effect during the PEO reaction, thereby affecting the conversion effect of the electric energy and the performance of the coating. Thus, limiting the leakage of electrical energy from the heterogeneous phase with weak film formation would be an effective method to improve the performance of PEO coating. For example, an additive capable of promoting heterogeneous phase film formation could effectively limit the leakage of electric energy, thereby improving the conversion efficiency of electric energy, ultimately improving the structure and performance of the coating.

## 4. Conclusions

A ceramic coating was prepared on an Al-Ti double-layer composite plate using PEO technology. The composite of Al and Ti did not affect the phase structure of the coatings on both surfaces of the composite plate. The P-Al coating was composed primarily of γ-Al_2_O_3_, and the P-Ti coating was composed of A-TiO_2_ and R-TiO_2_. Corresponding to the single metal, the thickness of the P-Al coating in the composite plate decreased, and the thickness of the P-Ti coating increased. The equivalent circuit of the Al-Ti coating connected the equivalent circuits of the P-Al and P-Ti coatings in parallel. During the PEO process, the electric energy was shifted toward the circuit with the lower blocking effect. Finally, the mechanism was extended to the PEO treatment on general metal matrix composites to broaden the application theory of the technology.

## Figures and Tables

**Figure 1 materials-12-00272-f001:**
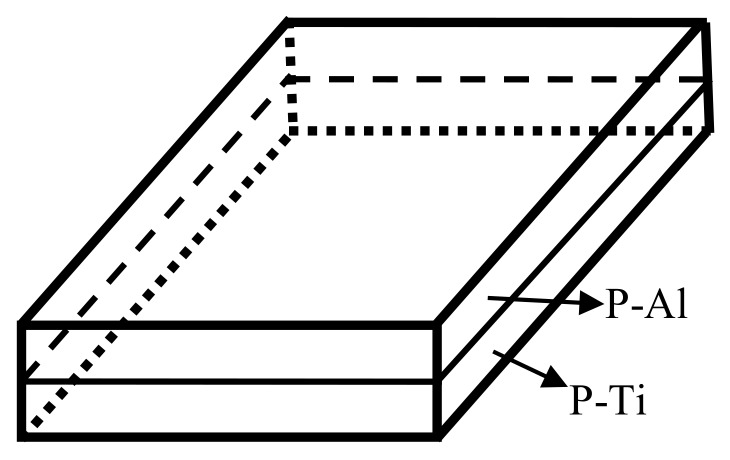
Schematic of the Al-Ti composite plate.

**Figure 2 materials-12-00272-f002:**
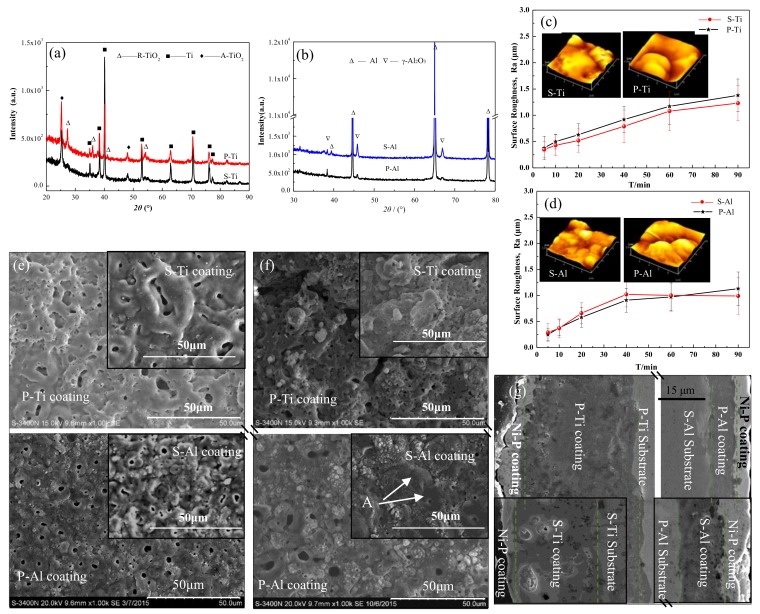
(**a,b**) X-ray diffraction (XRD) patterns of different samples; (**c,d**) the change in surface roughness (Ra) with treated time (insets are the atomic force microscopy (AFM) surface topography of different coatings treated for 60 min); **(e)** surface morphologies of Al-Ti coating treated 10 min; (**f**) surface morphologies of Al-Ti coating treated 10 min; (**g**) cross-section Al-Ti coating treated 60 min (insets are the coatings of the single metal).

**Figure 3 materials-12-00272-f003:**
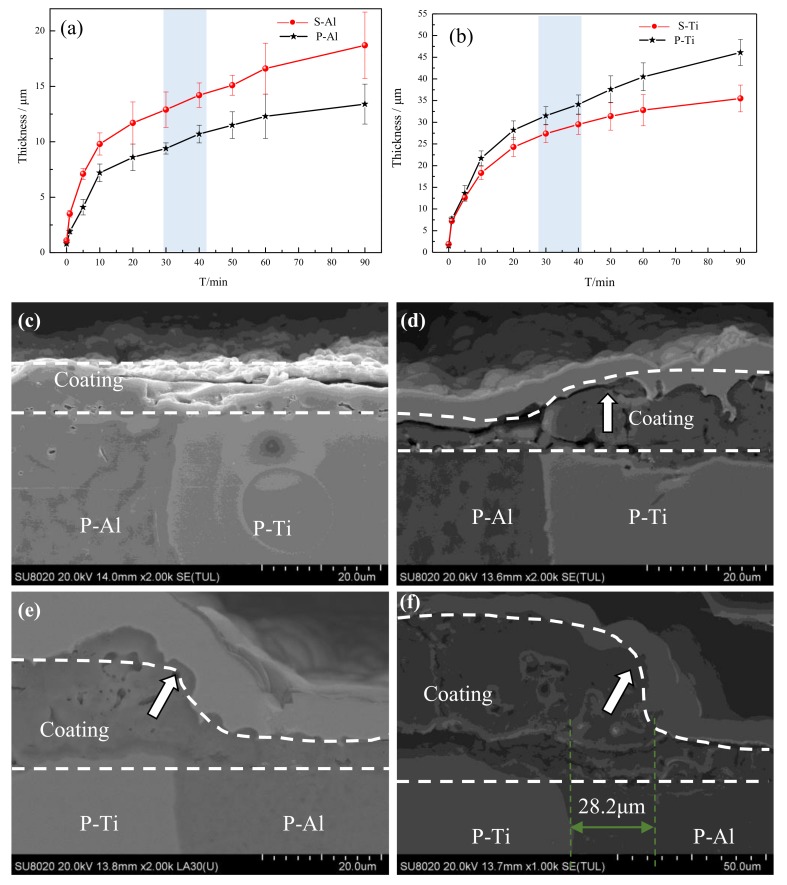
Dependence of thickness-time: (**a**) S-Al and P-Al coatings; (**b**) S-Ti and P-Ti coatings; growth process of coating on the welding joint: (**c**) 5 min; (**d**) 20 min; (**e**) 40 min; and (**f**) 90 min.

**Figure 4 materials-12-00272-f004:**
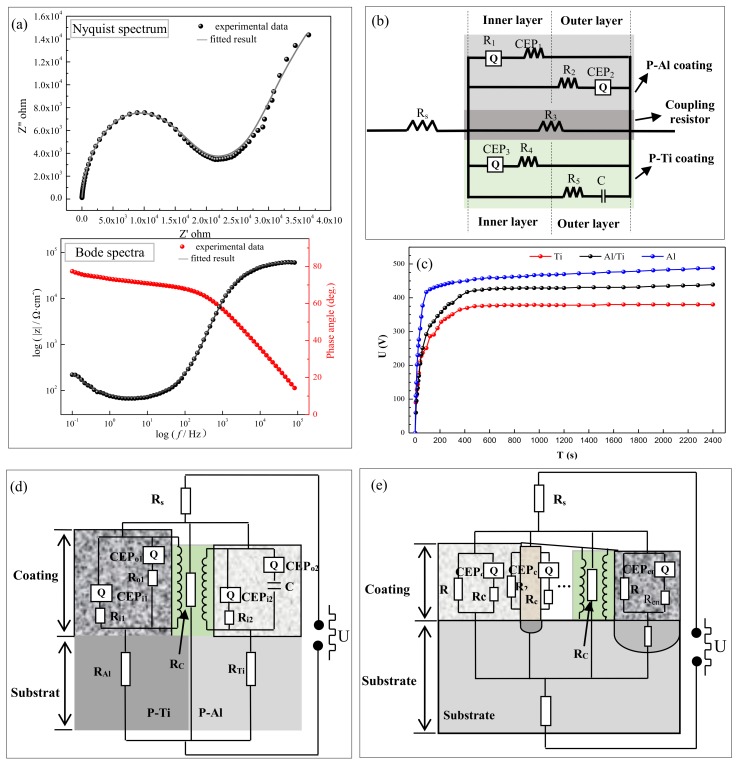
(**a**) Electrochemical impedance spectroscopy (EIS) spectra of the plasma electrolytic oxidation (PEO) coating at the welding joint; (**b**) equivalent circuit of the EIS spectra; (**c**) voltage-time curves at constant current mode of different samples; (**d**) schematic diagram of the equivalent circuit for the Al-Ti coating; (**e**) schematic diagram of the equivalent circuit for the general metal matrix composites (MMCs).

**Table 1 materials-12-00272-t001:** Values of electrical element of equivalent circuit.

R_s_ (Ω/cm^2^)	CPE_1_		R_1_ (Ω/cm^2^)	CPE_2_		R_2_ (Ω/cm^2^)	R_3_ (Ω/cm^2^)	CPE_3_		R_4_ (Ω/cm^2^)	C (F/cm^2^)	R_5_ (Ω/cm^2^)
	Q_1_ (F/cm^2^)	n_1_		Q_2_ (F/cm^2^)	n_2_			Q_2_ (F/cm^2^)	n_2_			
1.14	4.34 × 10^−^^8^	0.14	3.92 × 10^6^	1.61 × 10^−^^5^	0.93	1.15 × 10^6^	1.08 × 10^3^	2.02 × 10^−5^	0.96	8.69 × 10^4^	3.73 × 10^6^	8.83 × 10^12^

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
