# Peer review of "Investigation of Growth Mechanism of Plasma Electrolytic Oxidation Coating on Al-Ti Double-Layer Composite Plate"

_materials, 2019, doi:10.3390/ma12020272_

Reviewer 1 Report

1.      Spaces missing e.g.: line 31 (‘…Al alloy[1].’) also lines: 32, 34, 35, 37, 67… please check all the article carefully.

2.      Method names should be written in one of two possible ways (e.g.: Anodic Oxidation or anodic oxidation), where second one is preferred.

3.      Line 37: It is: ‘alloys[6, 7].’, should be: ‘alloys [6,7].’, please check all the article carefully.

4.      Lines 59-60: ‘Figure 1’ should be in one line.

5.      Line 64-65: Please provide finishing number of the papers.

6.      Please check the fonts, e.g.: line 72 – ‘40℃’, line 90: ‘(Figure 1(c1)),’. For b1, b2 etc. subscript instead of smaller font is suggested.

7.      Figure 1 b1, b2 – Y-axis description missing, also for text of the legend the font is too small.

8.      Figure 1: please improve the frames of all the pictures – should have the same color/thickness.

9.      Figure 3: please improve.

Author Response

Thank you very much for your letter and the comments concerning our manuscript entitled " Investigation of Growth Mechanism of Plasma Electrolytic Oxidation Coating on Al-Ti Double-layer Composite Plate "(ID: Materials-412599). These comments are all valuable and very helpful for revising and improving our paper, as well as important guiding significance to our researches. We have studied the comments carefully and have made corrections of the original manuscript. We hope all the mistakes were corrected and reasonable interpretation was done whenever required by the referees.

Sincerely yours,

The authors

All the questions are answered by the authors on a point-by-point basis. Revised portions are marked in red in the manuscript. Please see below the details of our replies to specific comments of the referees.

Reviewer 1#

1.Spaces missing e.g.: line 31 (‘…Al alloy[1].’) also lines: 32, 34, 35, 37, 67… please check all the article carefully.

Responds: Thank you for your careful reading of our manuscript. We have carefully corrected the writing, as shown on Lines 31, 32, 34, 35, 37, 67…

 2. Method names should be written in one of two possible ways (e.g.: Anodic Oxidation or anodic oxidation), where second one is preferred.

Responds: We are sorry for this writing mistake. We have carefully corrected this phrase according to your comment. Corrections could be found in line 34, 35...

3. Line 37: It is: ‘alloys[6, 7].’, should be: ‘alloys [6,7].’, please check all the article carefully.

Responds: Thank you for your careful reading of our manuscript. We have carefully corrected these writing mistakes, as shown on line 31, 32, 34, 35, 37, 67…

4. Lines 59-60: ‘Figure 1’ should be in one line.

Responds: Thank you for your careful work. We have improved all the Figure according to your comment, as shown on Figure 1, 2, 3, 4.

5. Line 64-65: Please provide finishing number of the papers.

Responds: I am very sorry that we did not pay attention to the detailed description of the sample pretreatment method. We have described in detail at manuscript, as shown on Lines 65-67.

6. Please check the fonts, e.g.: line 72-‘40℃’, line 90: ‘(Figure 1(c1)),’. For b1, b2 etc. subscript instead of smaller font is suggested.

Responds: Thank you for your reminding. I have made the corresponding revision at our manuscript, as shown on Lines 77…

7. Figure 1 b1, b2 --Y-axis description missing, also for text of the legend the font is too small.

Responds: Thank you for your reminding. I have made the corresponding revision at figure 2(a, b), as shown on Page 3.

8. Figure 1: please improve the frames of all the pictures -- should have the same color/thickness.

Responds: Thank you for your reminding. I have made the corresponding revision in all the figures.

9. Figure 3: please improve.

Responds: Thank you for your reminding. We have made improvements based on your comments, as shown on Page 7.

Reviewer 2 Report

Please find attached comments

Author Response

Thank you very much for your letter and the comments concerning our manuscript entitled " Investigation of Growth Mechanism of Plasma Electrolytic Oxidation Coating on Al-Ti Double-layer Composite Plate "(ID: Materials-412599). These comments are all valuable and very helpful for revising and improving our paper, as well as important guiding significance to our researches. We have studied the comments carefully and have made corrections of the original manuscript. We hope all the mistakes were corrected and reasonable interpretation was done whenever required by the referees.

Sincerely yours,

 The authors

 All the questions are answered by the authors on a point-by-point basis. Revised portions are marked in red in the manuscript. Please see below the details of our replies to specific comments of the referees.

Reviewer 2#

1. Although is far from my ability to correct the phrase construction, I recommend the manuscript to be revised by an English native. Some examples are:

(1) Line 88 “…Al and Ti didn't change the phase structure, which because…”

(2) Lines 92-93 “…but this phenomenon was no 92 found on P-Al coating…”

(3) Line 113 “…different samples were showed in Figure 2…”

Responds: Thank you for your careful work. For the problem of language writing, our manuscript has been revised by an English native.

2. Line 74, please define SEM the first time the acronym is used.

Responds: We are sorry for not addressing the various instruments used clearly. We have done the revision at manuscript, as shown on Lines 80-82.

3. Line 74, please specify the SEM equipment, model and the procedure to obtain the cross-sections that are present in figure 2. In fact, the “clean” cross-sections suggest that perhaps FIB was used. If so, please specify the conditions of FIB operation.

Responds: Thank you for your careful work. The "clean" cross-sections in our manuscript were obtained by the SEM, and the SEM equipment was specified at our revision, as shown on Lines 79-80.

Prior to SEM analysis, the samples were not prepared using FIB cutting techniques, but were manually ground and polished, and then washed with alcohol.

4. Line 86, please define A-TiO2 and R-TiO2.

Responds: Thank you for your careful work. The "R-TiO2" is the abbreviation of "Rutile TiO2", and "A-TiO2" is short for "Anatase TiO2", and we have done the revision at manuscript, as shown on Lines 98-99.

5. Line 109- Figure 1 caption, please specify in the caption b1, b2, c1 and c2, and not only generically as b) and c).

Responds: Thank you for your reminding. I have made the corresponding revision at all the figures, as shown on Page 2, 4, 5, 7.

6. Line 98, “…discharge pore aperture and higher roughness,…”, Please quantify some of the roughness parameters for all situations. It is not valuable to infer the higher or lower roughness just by morphological evaluation. Alternatively, remove the reference to the “higher roughness”.

Responds: Thank you for your comments. We have done supplementary experiments on the roughness of the different coatings. The surface roughness of the different coatings was tested by atomic force microscopy (AFM). Then, the roughness analysis was carried out by using the analysis software provide by the AFM equipment, and the average and standard deviation were calculated after the roughness measurement was performed in three different regions on the taken topographic map. The results have been done in our manuscript, as shown on Page 3.

7. Figure 2a1 and 2a2, Please specify in the materials section the number of measurements that were made in order to obtain the average and standard deviation bars of the thickness of the coatings. Also, specify if those measurements were all made in one sample or how many samples (replicates) were used for the measurements.

Responds: Thank you for your comments. During the experiment, five samples were taken for each parallel experiment to perform different tests, such as XRD, SEM, EIS, et al. We performed a simple thickness test after each sample was prepared and found that the thickness of each parallel sample was similar. Therefore, the measurement was performed 10 times at different places of the every sample, and determined the average value and its standard deviation. We have made a detailed explanation in our manuscript, as shown on Page 2.

8. Line 134, “…as shown of the arrow in Figure 3-9(c);”, Please correct the figure name and designation because figure 3-9 does not exist.

Responds: Thank you for your careful work. We have done the revision at manuscript, as shown on Page 7.

Reviewer 3 Report

In this paper, the authors attempt to demonstrate the growth using plasma electrolytic oxidation of a ceramic protection layer on a composite layer of Al/Ti. Although the authors demonstrate an example of the growth, the paper does not add sufficient improvement to existing methods and does not provide clear reasoning for the circuit deduction.

Specific Comments:

Capitalization of many words such as "Plasma" in the middle of a sentence is wrong.

Use of sub-figure labels should be (a), (b), (c), etc, and not (a1), (a2).

Acronyms are used without prior definition in the text such as "SEM".

When referencing an author, there is no need to put the first letter of the first name. "Xu et al. [9]" is fine.

No information is provided about how the thickness and dimensions of the samples were obtained.

How was the electrolyte temperature maintained? Was it a water bath or on a hot-plate?

Figure 1 is unclear. What is A and B? No description is provided for the inset.

Given the rough surface of the sample shown, the authors provide no indication how this affects the growth of the protection layer. A necessary experiment is to perform on smoother samples or comment how many samples were measured. Approximate roughness values extracted from AFM scans or measurement of grain sizes are essential to understand the growth mechanism.

How were the error-bars in Figure 2 obtained? Did the author do several measurements or were these just estimates?

Values for the circuit components the authors used to fit the experimental results should be provided.

The fitting of curve 3 (a) is not adequate. What was the R2 value?

Caption of Figure 3 is non consistent with the Figure.

How did the authors get from the circuit in 3 (b) to 3 (d), there might be a coupling resistor missing between the Ti and Al interface? And explanation must be given for 3 (e) as well.

Author Response

Thank you very much for your letter and the comments concerning our manuscript entitled " Investigation of Growth Mechanism of Plasma Electrolytic Oxidation Coating on Al-Ti Double-layer Composite Plate "(ID: Materials-412599). These comments are all valuable and very helpful for revising and improving our paper, as well as important guiding significance to our researches. We have studied the comments carefully and have made corrections of the original manuscript. We hope all the mistakes were corrected and reasonable interpretation was done whenever required by the referees.

Sincerely yours,

The authors

All the questions are answered by the authors on a point-by-point basis. Revised portions are marked in red in the manuscript. Please see below the details of our replies to specific comments of the referees.

Reviewer #3:

In this paper, the authors attempt to demonstrate the growth using plasma electrolytic oxidation of a ceramic protection layer on a composite layer of Al/Ti. Although the authors demonstrate an example of the growth, the paper does not add sufficient improvement to existing methods and does not provide clear reasoning for the circuit deduction.

Specific Comments:

1. Capitalization of many words such as "Plasma" in the middle of a sentence is wrong.

Responds: 

Thank you for your careful work. We have done the revision at manuscript, as shown on Lines 34-35.

2. Use of sub-figure labels should be (a), (b), (c), etc, and not (a1), (a2).

Responds: 

Thank you for your careful work. We have done the revision at manuscript, as shown in Figure 1, 2, 3, 4.

3. Acronyms are used without prior definition in the text such as "SEM".

Responds:

Thank you for pointing out our writing mistakes. We have done the revision at manuscript, as shown on Lines 79-80.

4. When referencing an author, there is no need to put the first letter of the first name. "Xu et al. [9]" is fine.

Responds:

Thank you for pointing out our mistakes. We have done the revision at manuscript, as shown on Line 40-41.

5. No information is provided about how the thickness and dimensions of the samples were obtained.

Responds:

Thank you for your reminder. The coating thickness was measured by a TT260B Coating Thickness Gauge. The measurement was performed 10 times at different places of the every sample, and determined the average value and its standard deviation. We have done the revision in our manuscript, as shown on Lines 86-88.

6. How was the electrolyte temperature maintained? Was it a water bath or on a hot-plate?

Responds:

Sorry, this was a mistake we made while describing the experimental process. The electrolyte temperature was kept below 40°C with water bath during the PEO process. We have made corresponding revised in our manuscript, as shown on lines 77-78.

7. Figure 1 is unclear. What is A and B? No description is provided for the inset.

Responds: 

Thank you for your reminder. We have deleted point B at the revision manuscript. And, the pint A referred to the plugging phenomenon of the discharge hole on the S-Al coating, as shown on Lines 120. 

8. Given the rough surface of the sample shown, the authors provide no indication how this affects the growth of the protection layer. A necessary experiment is to perform on smoother samples or comment how many samples were measured. Approximate roughness values extracted from AFM scans or measurement of grain sizes are essential to understand the growth mechanism.

Responds:

Thank you for your comments. We have done supplementary experiments on the roughness of the different coatings by AFM scan, shown as Lines80-84; and the results have been done in our manuscript, as shown on Page 3.

9. How were the error-bars in Figure 2 obtained? Did the author do several measurements or were these just estimates?

Responds: 

Thank you for your reminder. When testing the coating thickness, we performed 10 times at different positions on the surface of the sample, and then calculated the average value and the standard deviation, thereby obtaining the error-bars in Figure 3.

10. Values for the circuit components the authors used to fit the experimental results should be provided.

Responds:

Thank you for your reminder. The fitting results were showed in Table.1.

11. The fitting of curve 3 (a) is not adequate. What was the R2 value?

Responds:

Thank you for pointing out our mistakes. We have re-fitted the EIS data, and the equivalent circuit was showed in Figure 3. In addition, we have also analyzed the electronic components in the equivalent circuit, as shown on Page 5-6.

12. Caption of Figure 3 is non consistent with the Figure.

Responds:

Thank you for pointing out our mistakes. We have done the revision at Figure 3, as shown on Page 7.

13. How did the authors get from the circuit in 3(b) to 3(d), there might be a coupling resistor missing between the Ti and Al interface? And explanation must be given for 3 (e) as well.

Responds:

Thank you for your comments, a coupling resistor could exist between P-Al and P-Ti coating, we have done the analysis in our manuscript, as shown on page 7.

In order to explain the formation of the coupling resistance, we adjust the CE diagram to a physical diagram, that is, Figure 4(e). Thank you again.

Round  2

Reviewer 3 Report

The authors have made significant changes to the manuscript and answered my queries. The changes have resulted in a much improved paper that can not be accepted.